# Crop Improvement of *Moringa oleifera* L. through Genotype Screening for the Development of Clonal Propagation Techniques of High-Yielding Clones in Malaysia

**DOI:** 10.3390/biology13100785

**Published:** 2024-09-30

**Authors:** Sures Kumar Muniandi, Farah Fazwa Md Ariff, Mazura Md Pisar, Samsuri Toh Harun, Mohd Zaki Abdullah, Fauziah Abdullah, Siti Nur Aisyah Mohd Hashim, Syafiqah Nabilah Samsul Bahari, Norhayati Saffie

**Affiliations:** 1Forestry Biotechnology Division, Forest Research Institute Malaysia, Kepong 52109, Selangor, Malaysia; farah@frim.gov.my (F.F.M.A.); samsuri@frim.gov.my (S.T.H.); zaky@frim.gov.my (M.Z.A.); syafiqah@frim.gov.my (S.N.S.B.); norhayatisaffie@frim.gov.my (N.S.); 2Natural Product Division, Forest Research Institute Malaysia, Kepong 52109, Selangor, Malaysia; mazura@frim.gov.my (M.M.P.); fauziahabdullah@frim.gov.my (F.A.); sitinuraisyah@frim.gov.my (S.N.A.M.H.)

**Keywords:** moringa genotypes, superior clones, astragalin content, anti-inflammatory agents, anti-inflammatory bioassay, clonal propagation, yield improvement

## Abstract

**Simple Summary:**

*Moringa oleifera* L. is a multipurpose tree species widely used for its high nutritional value. It has also been used for centuries due to its high medicinal value and antifungal, antiviral, antidepressant, and anti-inflammatory properties, and has wider applications in the nutraceutical, pharmaceutical, and cosmetics industries. Due to greater demand in the local and international market, the exploitation of high-yielding moringa varieties and their cultivation techniques is needed. However, the breeding approaches adopted in Malaysia and achievements made for the exploitation of this crop were rather slow due to the unavailability of elite planting material and germplasm collection. Germplasm is a gene pool conserved in a wide array of crop species, landraces, and wild populations by identifying elite accessions with specific desirable genetic traits to increase yield and productivity in agricultural and horticultural crop species. Moringa germplasm collected across the country showed wider genetic variations for morphological traits and chemical content. Thus, moringa germplasm has been collected and conserved for the development of improved varieties with high-yielding potential for commercial cultivation in Malaysia. Hence, it is imperative to understand the genetic potential of each moringa genotype for the development of superior varieties through continuous breeding programs. Improvements in selection criteria, collection, and clonal propagation methods of moringa genotypes can result in the development of elite planting material with improved pharmacological and industrial properties and ensure the production of a high-quality end product.

**Abstract:**

*Moringa oleifera* L. is a valuable multipurpose tree species widely planted for centuries due to its high medicinal value and antifungal, antiviral, antidepressant, and anti-inflammatory properties in the food industry. However, its cultivation is hindered by production constraints such as the unavailability of planting material and the inadequate number of high-yielding clones. Thus, a study was initiated to select high-yielding clones in terms of growth and chemical content for the mass propagation of superior moringa trees. Screening on high-yielding clones with high astragalin content was conducted through the high-performance liquid chromatography (HPLC) analysis of moringa leaf extract. Selected genotypes were evaluated for their anti-inflammatory potential through in vitro bioactivity assays of leaf methanol extract. The effects of the rooting hormone, rooting substrates, and size of the cutting on the rooting response of branch cuttings of moringa were investigated. Results found that samples collected from different ecological zones of Peninsular Malaysia show significant variation in terms of astragalin content. The extracts were observed to show considerable variation in biological activity against the pro-inflammatory enzymes. The size of the cuttings had significant effects on the rooting of the cuttings as longer cuttings with bigger diameters rooted better than shorter cuttings with smaller diameters. Several genotypes of *M. oleifera* with superior phenotypic characteristics and bioactive compounds have been identified. Factors affecting the rooting efficiency and optimal conditions of rooting are suggested, which provides valuable information for the propagation of the superior planting material of moringa. This effort will ensure the sustainable production and supply of good quality raw materials for the production of quality end-products for the food and pharmaceutical industry.

## 1. Introduction

Since ancient times, useful plants with health-beneficial effects have been distinguished and traditionally used as medicine to treat various illnesses. Almost half a million plant species have been identified, cultivated, and used in traditional medicine all over the world [1]. This group of plants collectively known as medicinal plants can be considered the origin of modern medical technology. The trial-and-error method that has been practiced over the years from the beginning with knowledge, skill, and practices based on the theories, beliefs, and experiences in treating health issues has led to the development of plant-derived medicines [2]. The compound of plant origin has been used for centuries as one of the most important sources for drug manufacturing. It has been used not only to treat physical and mental illness but also for the maintenance of health, temporary relief from certain discomfort, prevention, and diagnosis [3]. With the proven side effects of expensive modern methods in treating illness, people have shifted towards low-cost, easily available traditional cures and treatments from plant-based medicine.

Plant species mainly from the family of Fabaceae or Leguminosae have been the source of most research in the search for a bioactive compound with excellent pharmacological properties. Considering the health benefits of the compound extracted from medicinal plants, many other plant species from various families were explored for their pharmacological properties. Moringaceae is a monogeneric family with a single genus moringa consisting of 13 other moringa species that is gaining worldwide recognition for having multipurpose uses and vast medicinal properties [4]. Most of the scientific reports from mainstream journals document the nutritive value and medicinal properties of *Moringa oleifera*, even though this species only represents 7% of the genus. The rest of the gene pool comprised 12 other species which have still not been fully explored and utilized, but they are equally as important and valuable [5]. 

*M.oleifera* L. is a multipurpose tree species commonly known as ‘drumstick tree’ or ‘horse-radish tree’ that originated from sub-Himalayan tracts of north-western India and has later been introduced into several countries of the tropic and subtropic region [6]. Moringa was first introduced as a feedstuff in the 1960s when various parts of moringa trees, such as leaves, seeds, and young branches, were used to feed farm animals. However, it gradually started to be used in human diets. Besides being a common dietary vegetable among people, it is also widely known for its health benefits. It is characterized by its wide application in the food industry because of its abundant nutritional ingredients, such as essential amino acids, oleic acid, vitamins, and minerals. It is considered one of the rare single plants that contain many essential nutrients in high quantities compared to those found individually in several different types of food and vegetables [7]. It has been used traditionally for centuries to treat multiple health disorders, such as anaemia, asthma, bronchitis, chest congestion, cholera, diabetes, and many other illnesses [4,7]. There is an abundant amount of scientific research documenting the excellent pharmacological properties of *M. oleifera* with antiviral, antimicrobial, anti-inflammatory, antioxidant, antidiabetic, antihypertensive, anti-tumour, anti-cancer, anti-ulcer, antipyretic, anti-epileptic, diuretic, and cholesterol-lowering agents [7,8,9,10,11]. Recent studies also indicate that *M. oleifera* isolate has the potential to prevent or treat COVID-19 or other coronavirus diseases, such as severe acute respiratory syndrome (SARS) and Middle East respiratory syndrome (MERS) [12]. *M. oleifera* has the potential application in the pharmaceutical industry due to its phytochemicals, such as phenolic acids, flavonoids, carotenoids, glucosinolates, and several other bioactive compounds. Excellent antioxidative and anti-inflammatory effects are attributed to moringa primarily because of its major phenolic compounds, such as lignans, 26 flavonoids, and 11 phenolic acids (and their derivatives). Some of the important flavonoids found in leaves are quercetin, kaemferol, apigenin, luteolin, and myricetin glycosides [13]. 

Astragalin (3-O-glucoside of kaempferol or kaempferol 3-glucoside) is one of the most therapeutically effective bioactive compounds found in moringa extract and is known for its wide pharmacological applications and for being an effective candidate in treating obesity because it promotes lipolysis and suppresses adipogenesis in 3T3-L1 adipocytes [14]. It is a naturally occurring flavonoid and therapeutically effective bioactive compound in most medicinal plants, and it is well known for its wide pharmacological applications and its antioxidant, anti-inflammatory, anticancer, antidiabetic, neuroprotective, cardioprotective, antiosteoporotic, antiulcer, and antifibrotic agents. Apart from its medicinal uses, *M. oleifera* extract has also been widely used in cosmetics as skin and hair care products; forage for livestock; coagulants; and biosorbents in water purification, phytoremediation, and biodiesel and bio-lubricant production [15,16,17,18]. Due to the broad spectrum of pharmacological features, this study aims to explore the genetic variability of moringa trees in terms of astragalin content and their anti-inflammatory activities for the selection of high-yielding varieties/genotypes.

Increasing health awareness among individuals on herbal supplements based on *M. oleifera* and the wide application of moringa extract for industrial uses has led to an upsurge in global market demand. It is estimated that the worth of the moringa ingredients market will increase from USD 5.5 billion in 2018 to over USD 10 billion by 2025 with the consumption of over 2.5 million tons [19]. With the ever-increasing global demand for moringa-based products and extracts, the gap between the demand and supply of *M. oleifera* raw materials is widening. The supply of raw materials to meet industry demands can be increased by raising intensively managed high-density *M. oleifera* plantations. The large-scale cultivation of *M. oleifera* trees with good silvicultural practices may be the solution in order to fulfil the gap between demand and supply. However, its cultivation for the sufficient supply of raw materials to the industry is hindered by production constraints, such as the unavailability of planting material and the inadequate number of high-yielding clones due to a lack of studies on the improvement and breeding of *M. oleifera*. The manipulation of provenance and genotype variation among trees can be used as a key factor in selecting superior mother trees with high-yielding ability, both in terms of vegetative growth and chemical content. The selection of clones with the high-yielding potential of astragalin content can improve the productivity of moringa trees in terms of the quality of the end product. 

*M. oleifera* can be propagated through direct seeding and cuttings and can be propagated throughout the year. Even though well-managed seed sources show a high rate of germination, moringa seeds generally exhibit late, erratic, and poor germination [20,21]. It is a common fact that seedlings raised from seed sources show a high level of phenotypic variation, and this can be a hindrance to the realization of true-to-type trees. *M. oleifera* has been successfully propagated experimentally through tissue culture, and a large number of published reports suggest the possibility of selected superior mother trees to be mass-produced for large-scale plantations [22,23]. However, in vitro propagation involves costly operational procedures and high-tech laboratory setups and equipment, which makes them out of the reach of rural people. To tackle the noted limitation, *M. oleifera* species are mainly propagated vegetatively using cuttings and, in some cases, air layering and grafting [24,25,26,27,28]. Propagation through the rooting of branch cuttings is therefore a better inexpensive option in producing true-to-type propagules with some modification carried out on a selection of appropriate cutting lengths and sizes, collection techniques, rooting media, and hormones, and maintenance of greenhouse conditions. This work is carried out to ascertain the role of tree selection and genotypes in producing a mother tree with high-yielding properties. Therefore, this research project aims to screen moringa genotypes based on their morphological characters, astragalin content, and anti-inflammatory properties for the identification of superior planting material. It was also designed to develop an inexpensive and efficient regeneration protocol for the propagation of selected genotypes of *M. oleifera* through branch cuttings for the development of high-yielding planting materials.

## 2. Materials and Methods

The experiments were conducted in the nursery of the Forest Research Institute of Malaysia, Kepong, Selangor. The study was conducted in an area that experiences a typically tropical hot humid climate with a 12 h day-and-night cycle. Extensive field visits were made to explore and identify potential mother trees for the collection of branch cuttings. The selection of mother trees was carried out at various sites from a few populations, covering the northern (Pulau Pinang), central (Negeri Sembilan), and southern parts (Johor) of Peninsular Malaysia based on some preliminary studies on the distribution and cultivation of *M. oleifera* trees in Malaysia. A total of 90 potential mother trees were selected for screening purposes (Figure 1).

### 2.1. Selection of Potential Mother Tree 

Potential mother trees were chosen from each region and marked on a global positioning system (GPS). The superior mother trees were selected based on their superior morphological characteristics. Mature trees with dense branching patterns that produce healthy vigorous branches and exhibit good growth potential with thick wide crowns and healthy green leaves, as well as resistance to disease and pest attacks, were selected for sample collection. Some of the important yield traits, such as pod size, pod length, the number of seeds per pod, and biomass, were also recorded for variety/cultivar selection. The selection was mainly performed on qualitative traits and some trees with good girth growth were also included in the selection criteria. However, selection based on height measurement is not applicable as most of the trees were planted for domestic purposes and trees were decapitated frequently at a certain height to enhance recovery rate, stimulate mass flowering, and produce more bushy leaves and pods within easy reach. 

### 2.2. Selection of Potential Mother Trees Based on Astragalin (AG) Content

A second screening was also carried out to select *M. oleifera* genotypes with the highest content of bioactive compounds, including astragalin (Figure 2). High-performance liquid chromatography (HPLC) analysis was carried out to evaluate the chemical content of selected *M. oleifera* genotypes. Leaves were collected from each mother tree, packed into plastic bags, and placed under room temperature. The leaf samples were cleaned and separated from stalk and debris before being dried in a hot air oven at 45 °C for three days. The dried leaves were ground into a fine powder and passed through a 0.45 µm laboratory test sieve. Samples were analysed using a high-performance liquid chromatography (HPLC) system (comprising a WATERS 2535 quaternary gradient pump, a WATERS 2707 autosampler, and a WATERS 2998 PDA), and a Luna C18 HPLC column (5 µm, 250 mm × 4.6 mm) with a gradient system consisting of 2 types of solvent denoted as A (0.1% formic acid in water) and B (acetonitrile). The chromatogram of the targeted compound was monitored at the wavelength of 340 nm. The genotypes with a high content of astragalin were screened and used as a starting material in the development of the propagation method. 

### 2.3. Anti-Inflammatory Bioassay of Moringa Genotypes with Chemicals Possessing the Highest Astragalin Content

Acetic acid, allopurinol, apigenin, bovine serum albumin (BSA), dimethyl sulfoxide (DMSO), hyaluronidase from bovine testes, hyaluronic acid, lipoxidase from glycine max (soybean), linoleic acid sodium salt, nordihydroguairetic acid (NDGA), potassium phosphate monobasic, sodium acetate trihydrate, sodium phosphate monobasic, xanthine oxidase from cow milk (ROCHE), and xanthine were of analytical grade and purchased locally from Sigma-Aldrich, St. Louis, MO, USA. The bioassays were performed using a BioTek Epoch 2 Microplate Spectrophotometer (Agilent, Santa Clara, CA 95051, USA).

#### 2.3.1. Lipoxygenase Inhibitory Assay (LOX)

The determination of lipoxygenase inhibitory activity was conducted using the spectrophotometer method, with minor modifications, as described by [29]. In the assay protocol, 160 µL of Na_2_H_2_PO_4_ buffer, 10 µL of test samples, and 20 µL of soybean lipoxygenase type-1B solution were mixed in a 96-well plate before being incubated in a spectrophotometer at 25 °C for 15 min. The enzyme reaction was initiated by the addition of 10 µL sodium linoleic acid (substrate) solution and incubated again at 25 °C for 10 min, and then absorbance measurements were taken at 234 nm. Nordihydroguaiaretic acid (NDGA) was used as a positive control.

#### 2.3.2. Xanthine Oxidase Inhibitory Assay (XO)

Xanthine oxidase inhibitory activity was measured by slightly modifying the spectrophotometric method [30]. Afterwards, 130 µL of potassium phosphate buffer (KH_2_PO_4_) (0.05 M, pH 7.5), 10 µL of the test solution, and 10 µL of xanthine oxidase solution were mixed and incubated for 15 min at 25 °C. The reaction was then initiated by the addition of 100 µL of the substrate in the form of xanthine solution and incubated again for 10 min at 25 °C. The enzymatic conversion of xanthine to form uric acid and hydrogen peroxide was measured at an absorbance of 295 nm. The performance of the assay was verified using allopurinol as a reference under the same experimental conditions.

#### 2.3.3. Hyaluronidase Inhibitory Assay (HYA)

The assay was performed according to the Sigma protocol with slight modifications [31]. The assay medium consisting of 1500–2000 U hyaluronidase in 100 µL of 20 mM sodium phosphate buffer was pre-incubated with 25 µL of the test sample (in DMSO) for 10 min at 37 °C. Then, the assay was commenced by adding 100 µL of hyaluronic acid and incubating it for a further 45 min at 37 °C. The undigested hyaluronic acid was precipitated with 1 mL of acid albumin solution. After standing at room temperature for 10 min, the absorbance of the reaction mixture was measured at 600 nm. Apigenin was used as a standard for verifying the performance of the assay under similar experimental conditions. 

### 2.4. Collection and Preparation of Branch Cuttings

Genotypes originating from the southern population TJM 23, TJM 24, TJM 27, and TJM 30 were mainly chosen for the mass propagation because of their high concentration of astragalin, considerable anti-inflammatory activity, and good growth characteristics both in the field and nursery. The hardwood part of the branch section was selected and harvested with a length range from 1 to 1.5 m using a handsaw. The greenish part of the section was discarded as they were too young to produce roots under nursery conditions. Selected branch segments were gathered together and kept under shade conditions for 2 days as they were prepared for rooting. Before treatment and planting, the branch section was segmented into smaller cutting lengths ranging between 30 to 60 cm according to the required length for each experiment. Uniform-sized cuttings consisting of at least three to four active nodes were chosen to avoid non-treatment variations. Segmented cuttings were washed with running tap water and brushed if needed to remove impurities attached to the surface before being pre-treated with fungicide. A fungicide solution with 1% benomyl was prepared in distilled water and cuttings were immersed for 1 to 2 min. The branch cuttings were air-dried for a few minutes before being transplanted to appropriate rooting substrates. 

### 2.5. Effect of Hormone Treatment on Rooting Ability and Survival of M. oleifera Cuttings

The effects of rooting hormones on the rooting ability of cuttings were studied by treating cuttings with 0 (control) (T1), 1000 ppm (T2), 3000 ppm (T3), and 5000 ppm(T4) of indole-3-butyric acid (IBA) solution for a few seconds. IBA formulations were prepared by dissolving the IBA powder in absolute ethanol and diluting the dissolved solution in varying amounts of distilled water to obtain the required concentrations. Prepared cuttings were transplanted directly on moist river sand. Thirty uniform cuttings with lengths of approximately 40 to 45 cm and girths ranging from 40 to 45 mm were planted for each treatment.

### 2.6. Effects of Cutting Length and Size on Rooting Ability and Survival of M. oleifera Cuttings

Based on the results obtained in the above experiment, a second experiment was designed using cuttings of various lengths and sizes. To study the effects of cutting lengths and girths on the rooting ability of *M. oleifera* cuttings, (i) branch cuttings were sectioned into three different lengths (30 cm (L0), 40 cm (L1), 50 cm (L2), and 60 cm (L3)) with a uniform girth of approximately around 60 mm; (ii) branch cuttings were sectioned with a uniform length of approximately 60 cm but with different girth sizes of 2.5–≤3.5 cm (D0), 3.5–≤4.5 cm (D1), 4.5–≤5.5 cm (D2), and 5.5–≤6.5 cm (D3). The sectioned cuttings were pre-treated with the same procedure mentioned in Section 2.4 before being transplanted to polyethylene bags containing rooting substrates with a mixture of topsoil, sand, and compost (2:3:1). Thirty cuttings were used for each treatment without any incorporation of rooting hormones. 

### 2.7. Effects of Rooting Substrates on Rooting Ability and Survival of M. oleifera Cuttings

An experiment was designed to investigate the possibility of enhancing the rooting ability with different types of rooting substrates. Prepared *M. oleifera* cuttings were transplanted to polyethylene bags containing different types of rooting substrates without any treatment of the rooting hormone. Thirty cuttings of uniform length and size, approximately 60 cm in length and 60 mm in girth, were used for each treatment. The rooting substrates prepared for this experiment were sand (M1); a mixture of topsoil, sand, and compost (3:2:1) (M2); a mixture of topsoil, sand, and compost (2:3:1) (M3); a mixture of topsoil, sand, and burnt rice husk (2:3:1) (M4); and a mixture of topsoil, sand, and coco peat (2:3:1) (M5).

### 2.8. Growth Conditions and Observations

Prepared cuttings (one-third of the length) were placed vertically into the rooting substrate. The transplanted cuttings were placed in the shade, allowing 50% sunlight in the greenhouse equipped with an automated irrigation system. Cuttings were allowed to grow with constant monitoring for 3 months. Observations were made from time to time to investigate the survivability, bud break, and shoot growth of the cuttings.

### 2.9. Acclimatisation of Rooted Cuttings in the Nursery

After 3 months of setting the experiment, rooted cuttings were transferred into polythene bags filled with topsoil, sand, and compost (2:3:1) kept in shade for one week, and relocated to an open area with full sunlight for several months to assess the initial performance, survival, and growth of rooted cuttings. 

### 2.10. Growth Performance of Clones in the Field under the Same Environmental Conditions

The growth of each clone was planted under the same environmental conditions for the evaluation of growth stability and the further screening of superior mother trees. Selected clones were planted at the Forest Research Institute Malaysia (FRIM) research station in Maran, Pahang. A total of 120 trees (3 sources × 12 genotypes × 4 trees for each genotype) were planted for initial screening purposes. The growth performance of each clone was monitored and irrigated regularly with a consistent application of fertilizer NPK every month. 

### 2.11. Data Collection and Analysis

After 3 months under nursery conditions, the cuttings were removed from the substrates and washed to remove any remaining soil attached to the roots for measurement purposes. Various parameters, such as the survival rate, rooting percentage, number, and length of roots; the number and length of shoots per cuttings; and the fresh and dry weight of roots and shoots, were calculated. The lengths of the shoots and roots of cuttings were measured from the collar region to the shoot or root tip and expressed in centimetres. For the calculation of the dry weight of shoots and roots of each cutting, the samples were placed in paper bags and oven-dried separately at 85 °C for 48 h (root) and 24 h (shoot). The root parts of the cuttings were removed from the cuttings and cleaned thoroughly with running tap water before being placed in the oven. Fresh weight and dry weight were taken before and after the drying procedure and expressed as grams per cutting. The observed data were analysed with Statistical Package for Social Sciences (SPSS) (Version 22) to assess the possible variations in treatments at *p* ≤ 0.05. The mean separation test was conducted using the Duncan Multiple Range Test (DMRT) to compare the mean ANOVA values among the treatments. For the anti-inflammatory bioassay, all analysis was carried out in triplicate and experimental results were expressed as mean ± standard errors. A dose-dependent study of the tested samples was reported as IC50 values as a measure of the potency of a given sample and compared with a positive control by plotting percentage inhibition as a function of inhibitor concentration and fitting in a normalized dose–response non-linear regression curve using GraphPad Prism 8. Statistical differences between samples were analysed using one-way analysis of variance (ANOVA) followed by Tukey’s multiple comparison test for mean separation. The mean values were considered statistically significant if *p* < 0.05. Pearson’s correlation, a well-established statistic for calculating the association between two variables, was applied to measure the co-expression of astragalin content and the IC50 of biological enzyme inhibition. 

## 3. Results and Discussion

### 3.1. Genotype Screening and Quantification of Astragalin (3-O-Glucoside of Kaempferol) Content in Selected Moringa oleifera Mother Trees

The health benefits of *M. oleifera* leaves are attributed to phytochemical constituents that have physiological effects on the human body. Extensive phytochemical evaluations on *M. oleifera* have shown the presence of various phytoconstituents and compounds, including flavonoids, phenolics, alkaloids, and fatty acids. These constituents are believed to exert protective as well as beneficial effects on multiple disease states, including inflammatory-related illnesses, cancer, and cardiovascular diseases [32,33,34,35,36]. One of the most active major compounds that has been linked to all the biological activities of moringa extract, especially as an anti-inflammatory compound, is astragalin. In this study, the determination of astragalin content in each selected genotype was conducted as an initial stage for the genotype screening and identification of superior genotypes based on their bioactivity (Figure 3). This initial selection intends to evaluate the relationship between chemical content and the superior morphological variation of each genotype. Genotypes were selected and vegetatively propagated for the establishment of germplasm plots at different environmental conditions to evaluate their stability and adaptation in terms of their vegetative growth, biomass production, and chemical content. The highest concentration of astragalin was found in samples from the southern part (TJM) and samples from the northern part (NSM) exhibited the lowest astragalin content (Figure 4). The variation in astragalin content may be caused by the different environmental factors from where the samples are being collected. It was found that the concentration of major active compounds in *M. oleifera* leaf extract varies according to population, season, and environmental conditions from where the samples were collected [37,38]. In accordance with the study, samples collected from cooler regions resulted in the highest content of astragalin compared to samples originating from warmer regions in this study. Thus, further studies need to be carried out to assess the effect of genetics and the environment on the production of astragalin from these genotypes. Their performance needs to be tested in the same environmental conditions for the evaluation of genetic stability and adaptation. Apart from the donor plant, the maturity stage, agro-climatic conditions, collection time, and extraction methods are some of the other determining factors that could also influence the phytochemical production in a certain plant species [38,39].

### 3.2. Anti-Inflammatory Bioassay of Selected Moringa oleifera Genotypes

In our search for prospective anti-inflammatory agents derived from natural resources, ten extracts prepared from the leaves of *M. oleifera* collected from ten selected individuals/genotypes (Table 1) were screened for their inhibitory properties. In this study, pro-inflammatory enzymes, namely hyaluronidase, lipoxygenase, and xanthine oxidase, were used as in vitro bioassays to assess the anti-inflammatory properties of *M. oleifera* leaf methanol extracts. The half-maximal inhibitory values (IC50) of all the extracts are tabulated in Table 2.

The extracts were observed to show considerable variations in biological activity against the pro-inflammatory enzymes. The IC50 value signifies the amount of sample concentration required for lowering the enzyme’s activity by half. A lower value indicates a more potent inhibitor for a specific assay, as a lower concentration is required to attain the same level of inhibition. Among the 10 samples tested for lipoxygenase inhibitory potential, the IC50 values of the sample coded as TJM30 had the lowest value of 245.67 ± 2.42 µg/mL. The IC50 values of the remaining samples were found to be between 307.37 5.71 µg/mL and 423.43 4.36 µg/mL. In the xanthine oxidase inhibitory assay, the results showed low inhibitory activities for the reference standard allopurinol with IC50 values for some of the extracts ranging from 336.60 ± 174.52 to above 1000 µg/mL, making it the least potential source for xanthine oxidase inhibitors. 

The hyaluronidase inhibitory activities ranged from 108.96 ± 6.15 to 633.65 ± 59.81 µg/mL for the tested extracts. The sample NSM16 showed the most profound activities with the lowest IC50 value followed by NSM 22 and TJM 23 when compared to the reference standard (Table 2; apigenin IC50: 100.59 ± 5.70 µg/mL). The hyaluronidase inhibitory property of these extracts was recorded for the first time to upgrade the medicinal value of this species. *M. oleifera* extracts show modest anti-inflammatory effects in vitro in terms of lipoxygenase inhibitory activity, and they are a prospective candidates as hyaluronidase inhibitors. The differences in the impact of different extracts on anti-inflammatory activity in the current study can be explained by the variation of bioactive groups present in the extract. The many active biological properties of this species are thought to derive from the secondary metabolites in its matrix. The presence of high content of phenolic compounds, especially astragalin, justifies the anti-inflammatory activities of *M. oleifera*. A Pearson correlation coefficient was used to express the relationship between the amount of astragalin and the IC50 of biological enzyme inhibition (Table 3). While lipoxygenase had a positive association with astragalin levels, the inhibition values of hyaluronidase and xanthine oxidase demonstrated a reverse correlation. These associations, however, were deemed weak and not statistically significant between the level of astragalin in each extract and the bioactivity analysis. This implies that, in addition to astragalin, other chemicals or action synergisms may also be connected to the reported activity. The correlation relationship between astragalin and anti-inflammatory biological activities is shown in Table 3 and Figure 5.

### 3.3. Development of Vegetative Propagation Techniques through Branch Cuttings of Selected Moringa oleifera Genotypes

The rooting percentage of *M. oleifera* branch cuttings was significantly affected by the applied IBA concentration. Cuttings without the application of hormones in the control treatment produced the highest rooting percentage (80.0%) and showed no significant difference with cuttings treated with 1000 ppm of IBA. An increase in IBA concentration negatively affected the rooting percentage of moringa cuttings and recorded the lowest percentage in T3 (28.0%). Growth parameters like the number and length of the shoots and the number of roots were all recorded to be non-significantly affected by the different concentrations of IBA (Table 4). The cuttings were able to initiate multiple shoots and roots with a high rooting percentage, even without the treatment of the rooting hormone IBA hormone. The cuttings were able to produce an average of 6 to 8 healthy shoots from the active buds after only 2 weeks in nursery conditions (Figure 6). However, not all cuttings with newly developed shoots successfully initiated root and thus were not able to survive for more than 3 months. The leaves became yellow, wilted, and dry, which eventually caused the whole cuttings to rot in less than a week and die.

Even though the cuttings were able to initiate roots without any hormone treatment, the rooting ability in terms of root length was significantly enhanced through the incorporation of 1000 ppm IBA. A high mortality rate was observed among cuttings treated with a concentration above 3000 ppm. The longest length of the root was recorded in cuttings treated with 1000 ppm IBA (20.5 cm) followed by cuttings in the control treatment (8.8 cm). Cuttings without hormone treatment were found to develop a more enhanced vegetative shoot growth compared to the IBA-applied treatments. Most of the cuttings treated with an IBA concentration of 5000 ppm produced fewer tiny roots (Figure 7). The calculation of fresh and dry weight was based on the accumulation of total vegetative parts harvested from the cuttings. Thus, it was expected that, in most instances, the higher the number of long roots produced per cutting, the higher the value of fresh and dry weight. Cuttings treated with 1000 ppm IBA possess a significantly higher mean value of root length recorded with equally higher values of fresh and dry weight. The same relationship could not be observed in the other treatments. 

The findings from this experiment support the theory that the pre-treatment of cuttings with exogenous rooting hormones is not essential in the propagation of *M. oleifera*. Rooting hormones, collectively called auxins, can exert influence on polysaccharide hydrolysis, which results in the increment of physiologically active sugar in the cuttings. This will eventually provide more energy and stimulate more root formation from meristematic tissues and root primordia [40,41]. Auxins are produced from vegetative parts, such as leaves and buds, and travel down through the phloem with essential nutrients to be concentrated in areas requiring root stimulation [26]. Thus, the rooting hormone has been commonly applied exogenously to the branch cuttings of various species to stimulate root formation and growth. The rooting response to the hormone depends on the content of endogenous auxins in each species, which varies due to the age factor, physiological conditions, the genetic constituent of a mother plant, and the time of year [42]. The application of exogenous synthetic rooting hormones could promote a proper hormonal balance that favours rooting from the cutting if the endogenous auxin content is insufficient. However, the high content of naturally occurring secondary metabolites in the branch cuttings of moringa trees may have assisted in the formation of roots from non-treated cuttings and resulted in non-significant variations among the treatments [26,43].

Some related studies also highlighted the non-significant role of IBA treatment in enhancing the rooting ability of cuttings [26,44]. Rufai et al. [26] found that the incorporation of exogenous IBA at various concentrations non-significantly affected the number of primary branches, Spad chlorophyll content, plant height, root number, and root length of *M. oleifera* cuttings. In some instances, the control even outperformed the IBA-treated cuttings. Other than IBA, other auxins such as naphthalene acetic acid (NAA) and kinetin were also found significantly affect the rooting ability of cuttings of various species. Further studies on the application of one of these hormones, or in a combination of hormones, could assist in better elucidating the role of the rooting hormone in enhancing the rooting ability of moringa cuttings. Apart from the growth hormone, other influencing factors determining the rooting ability of cuttings, such as the size of the cuttings and growth substrates, need to be investigated to further enhance the propagation techniques of *M. oleifera*. Thus, the growth and development of moringa cuttings were observed in subsequent experiments utilizing cuttings of various sizes and in different rooting substrates.

The rooting ability was discovered to be significantly affected by an increment in the size of stem diameter with regards to the number of shoots and roots, the root length and number, and the shoot and root dry weight (Table 5). The highest rooting percentage was observed in smaller cuttings with a size of 2.5–≤3.5 cm (D0) followed by cuttings in other treatments. Cuttings in treatments D1, D2, and D3 were not significantly different in terms of rooting percentages; however, cuttings from D3 were found to produce significantly more shoots and roots with a greater shoot and root dry weight. Cuttings in treatment D3 produced the highest number of shoots (9.83) and roots (7.83 cm) per cutting but took a long time to produce roots compared to thinner cuttings. Thinner cuttings were rooted much faster than the thicker cutting; however, they easily became stunted with a smaller number of shoots. A high mortality rate was observed among cuttings with a smaller diameter after 4 months in nursery conditions.

From the results obtained, the increased diameter of the stem significantly increases other vegetative growth traits, such as the shoot number, shoot length, root number, root length, and dry weight of shoots and roots produced per cutting. Secondary growth generated by the cell division of the vascular cambium between the xylem and phloem tissues increases the stem diameter in annual plants and intensifies the vigorous formation of shoots and leaves from the stem [26]. The effect of storage capacity in cuttings with the largest diameter may be more critical than length [45]. This relationship between rooting and the size of the cuttings could be associated with an increase in cutting volume (length x diameter). Larger cuttings tend to have better rooting abilities compared to smaller cuttings due to their intrinsic factors. The larger cuttings store more energy reserves in the form of carbohydrates, with high levels of endogenous auxins and other root-inducing factors for the improved formation of roots from the base of the cuttings [45]. Root formation from cuttings is assisted by the retention of leaves on cutting through post-severance carbon assimilation [46]. However, no leaves were retained on the moringa cuttings in our experiment. Thus, an increase in the rooting percentage with an increase in cutting size may be caused by the high content of carbohydrates in the larger cuttings. Thicker cuttings can also be associated with the age or maturity of the branch section. Stems and branches are oldest and most mature in terms of ontogenetic age; thus, field sources consisting of different maturity levels and ages are difficult to propagate via stem cutting and show a very poor rooting ability compared to juvenile or rejuvenated materials [47]. However, mature branches were found to be more suitable for the propagation of moringa cuttings compared to juvenile materials. 

Based on the ANOVA, cutting length had no significant effects on the rooting percentage and shoot length of moringa cuttings; however, it significantly affected the number of roots produced per cuttings, as well as the shoot and root dry weight (Table 6). The highest shoot number (6.23) and root number (6.27 cm) were observed in longer cuttings, i.e., 60 cm (L3), followed by cuttings in treatments L2 and L1. Shorter cuttings were found to be equally good in producing root in terms of rooting percentage, and the longest root length of 9.2 cm was recorded in treatment L0 (30 cm). Similar to the shooting response observed in cuttings with various diameters, shorter cuttings produce a significantly smaller number of shoots, which may be due to the existence of a smaller number of active nodes compared to the longer ones. Cutting length is involved in water velocity and nutrient transfer. Reduced rates of acropetal transport of cytokinin and basipetal transport of auxin could hinder the development of adventitious root primordia. The ratio of both growth hormones should be high in the stem base for the development of root primordia [45,48,49]. 

Cuttings with diameters ≥ 5.0–6.0 cm and lengths ≥ 50–60 cm gave better rooting values compared to most of the vegetative parameters tested. Cuttings with smaller diameters and shorter lengths resulted in higher mortality rates, even after the formation of vigorous shoot growth after 2 months in nursery conditions. It was observed that the unrooted cuttings or cuttings with poor root formation eventually shed their leaves and wilted when their carbohydrate reserves were depleted. However, the relationship between the rooting ability and cutting diameter was inconsistent and greatly depended on the species, age, and condition of the stock plant [50]. The negative relationship might be due to the inability of the plant to convert the reserve starch in the cutting to sugar and a greater extent of dependency on current assimilates rather than carbohydrate reserves [50,51]. Other than that, the secondary growth and thickening of the lignin layer in cuttings with bigger diameters may create a physical barrier to root formation [52]. The rooting ability of branch cuttings mostly depends on many plants’ abilities to control the interaction of internal factors, such as nutrition and hormone and external factors (including the inherent capacity of the plants to form rooting primordia and roots in cuttings) [53]. Some species can readily form roots, while others may be very difficult to form roots.

In terms of different types of rooting substrates, significant effects were observed in rooting percentages and most of the rooting parameters tested in the study (Table 7). Treatment M1 with only sand as the rooting substrate was found to result in the highest mean rooting percentage (76.7%), the number of shoots per cutting (10.9), shoot dry weight (37.9 g), and root dry weight (3.02 g). The shoots and survival rates of cuttings grown in substrates containing more sand (M2) were also better compared to cuttings grown in other substrates. The highest mean number of roots produced per cutting was obtained in treatments M1, M2, and M3, followed by M4 and M5, whereas the highest mean root length was recorded in treatment M3 (8.52 cm) (Figure 8). It has been found that cuttings of some species show some preference for rooting substrates due to the differences in the components of the substrates, nutrient content, aeration, and moisture-holding abilities. This could be related to the good aeration system and increased respiration rates at the basal end of the cuttings grown in sand substrates, providing a conducive environment for root growth and development [25]. Water uptake using cuttings, which is related to the volumetric water content of the substrate, reduces water deficit and enhances the rooting ability of cuttings. Water uptake using cuttings is essential in order to overcome the initial physiological shock when the cutting is taken from the mother plant. However, excess water intake in moringa cutting leads to high moisture surplus and death of the cutting. Irrigation frequency and high moisture content trapped in the substrates containing coco peat may contribute to the low survivability of cuttings grown in M5. *M. oleifera* cuttings are observed to have good growth under good irrigation systems; however, a high amount of water supply has inhibitory effects on root growth and root rotting. 

## 4. Conclusions

Several genotypes of *M. oleifera* with superior phenotypic characteristics and bioactive compounds (including astragalin) were identified. The results encourage the further investigation of in vitro and in vivo studies, along with detailed phytochemical investigations, so that we can potentially use these genotypes in the prevention and therapies of inflammatory-related diseases. The production of planting materials in the nursery is possible through clonal cuttings for the establishment of breeding stock plots at multiple locations. *M.oleifera* could be considered a ready-to-root species since most of the cuttings produced roots even without the application of rooting hormones. Three-to-four-node cuttings, 60 cm in length and 6.0 mm in diameter, were amenable to cloning with or without auxin treatment. However, the growth and development of the rooted cutting were further enhanced with the manipulation of the size of the branch cuttings, as well as the types of rooting substrates, increasing the survival rate of cutting raised in a nursery. Improvements in the selection criteria, collection, and clonal propagation method of *M. oleifera* genotypes can result in the production of improved planting material with high yields for the food and pharmaceutical industry.

## Figures and Tables

**Figure 1 biology-13-00785-f001:**
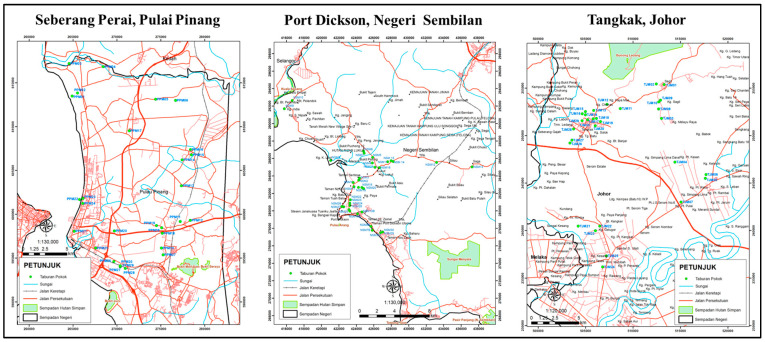
Green dots indicate the distribution of selected mother trees of *Moringa oleifera* in Peninsular Malaysia.

**Figure 2 biology-13-00785-f002:**
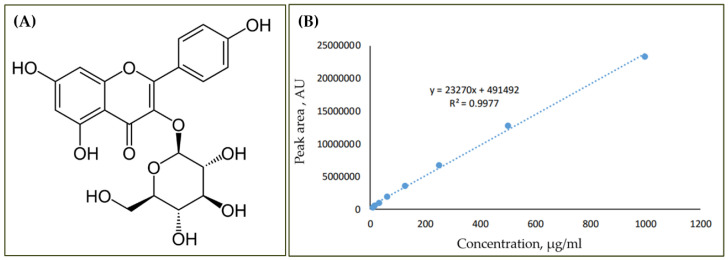
(**A**) Chemical structures of isolated compound astragalin (AG) and (**B**) plot showing the calibration curve of standards.

**Figure 3 biology-13-00785-f003:**
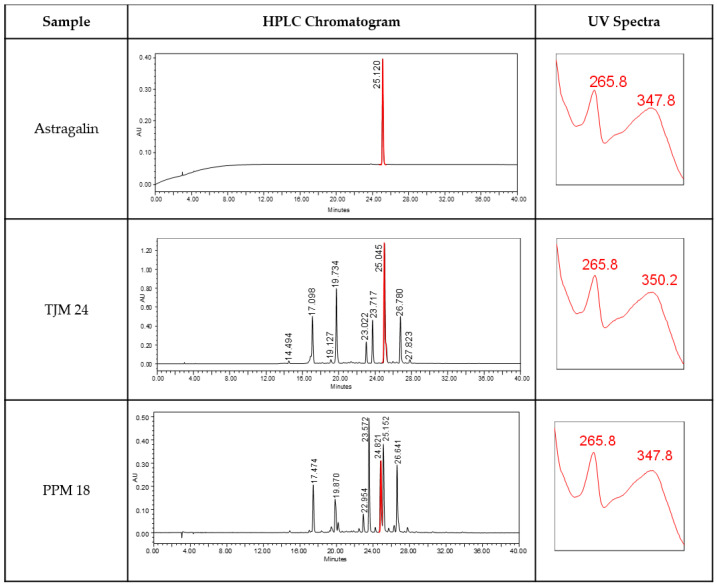
HPLC chromatogram of astragalin in representative samples of *Moringa oleifera* genotypes at 340 nm. Abbreviation: AU, absorbance units; red peak represent astragalin in samples.

**Figure 4 biology-13-00785-f004:**
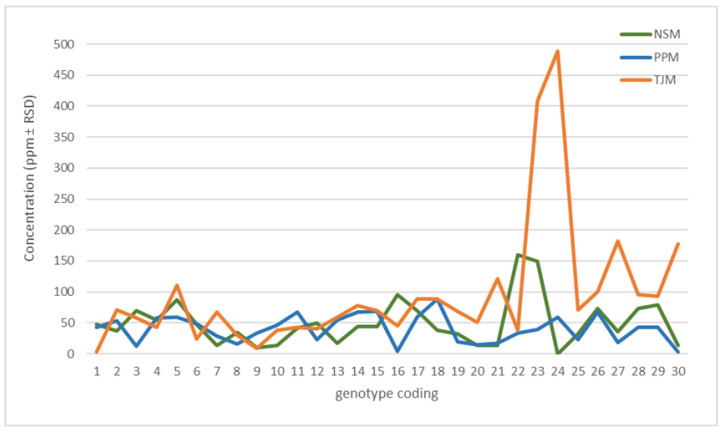
The average concentration of astragalin in *Moringa oleifera* leaf samples collected from three populations shows significant variation among genotypes. (NSM = central region; PPM = northern region; TJM = southern region.)

**Figure 5 biology-13-00785-f005:**
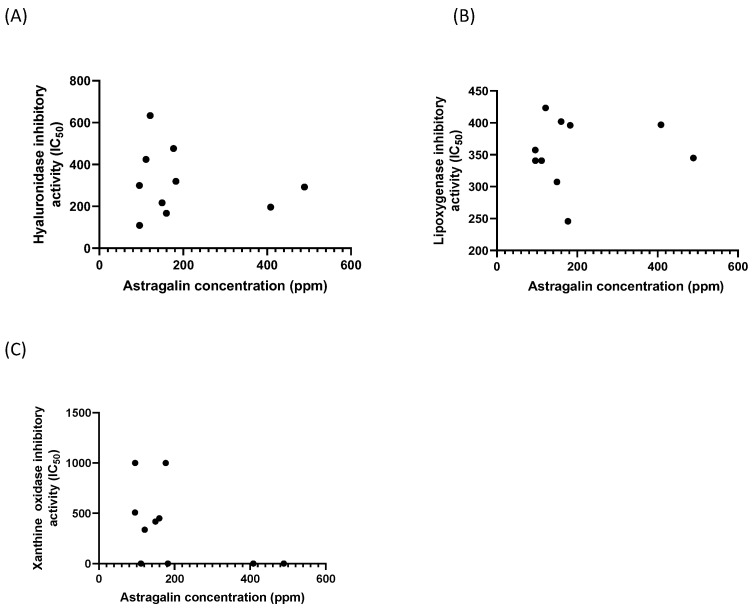
Pearson correlation scatter plot showing relationships between (**A**) astragalin and hyaluronidase, (**B**) astragalin and lipoxygenase, and (**C**) astragalin and xanthine oxidase.

**Figure 6 biology-13-00785-f006:**
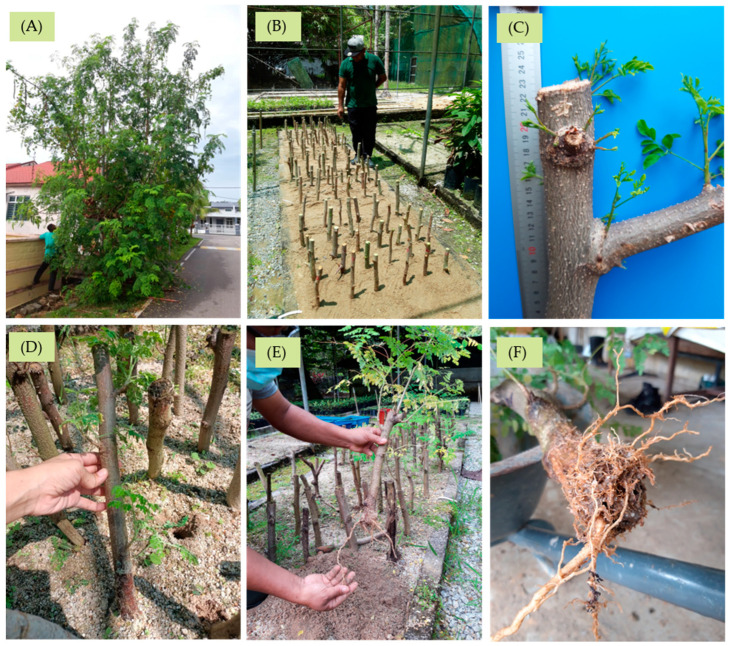
(**A**) Plus tree selection and collection of planting material of *Moringa oleifera* for propagation, (**B**) preparation of sand beds and planting of indole-3-butyric acid (IBA)-treated cuttings under nursery conditions, (**C**) vigorous shoot growth of cuttings after 2 weeks under nursery conditions, (**D**) bud break and vigorous shoot growth from treated cuttings without any formation of root after 3 months under nursery conditions, and (**E**,**F**) healthy shoot and root growth from cuttings treated with 1000 ppm IBA after 3 months under nursery conditions.

**Figure 7 biology-13-00785-f007:**
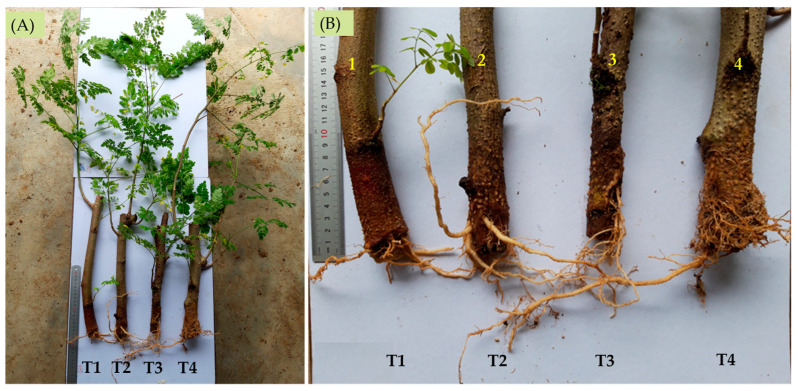
(**A**) Shoot growth and (**B**) root growth of *Moringa oleifera* branch cuttings in different concentrations of indole-3-butyric acid (IBA) solution, T1 (control), T2 (1000 ppm), T3 (3000 ppm), and T4 (5000 ppm) after 3 months under nursery conditions.

**Figure 8 biology-13-00785-f008:**
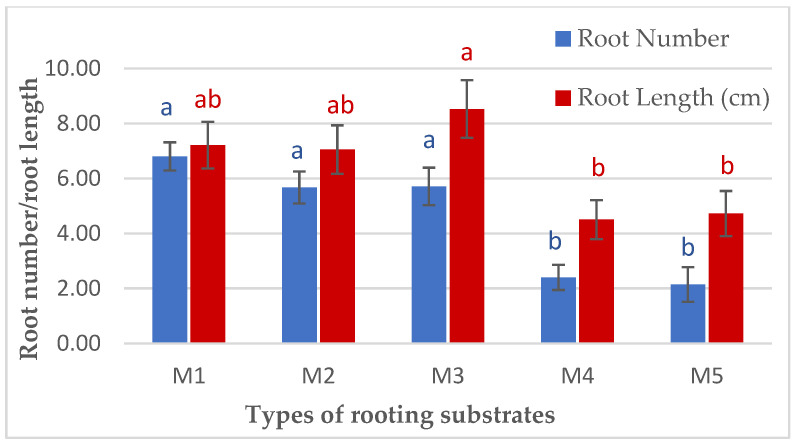
Number and length of roots produced from branch cuttings of *Moringa oleifera* in different types of rooting substrates. Mean values are expressed as ± standard errors of the mean, and the means of each variable with the same letters are not significantly different at *p* ≤ 0.05.

**Table 1 biology-13-00785-t001:** The selection of the ten best *Moringa oleifera* genotypes with high concentrations of astragalin content from three regions. TJM refers to samples collected from the southern region and NSM refers to samples from the northern region of Peninsular Malaysia.

(Samples)	Average Concentration of Astragalin ± RSD (ppm)	Average Percentage of Astragalin in the Sample ± RSD (*w*/*w*)
TJM 24	489.26 ± 0.61	0.49 ± 0.61
TJM 23	408.40 ± 2.30	0.41 ± 2.30
TJM 27	182.57 ± 1.86	0.18 ± 1.86
TJM 30	176.93 ± 0.56	0.18 ± 0.56
NSM 22	159.89 ± 1.86	0.16 ± 1.86
NSM 23	149.54 ± 0.50	0.15 ± 0.50
TJM 21	121.32 ± 0.60	0.12 ± 0.57
TJM 5	111.08 ± 2.65	0.11 ± 2.65
TJM 26	99.99 ± 0.30	0.10 ± 0.30
NSM 16	95.81 ± 1.38	0.10 ± 1.38

**Table 2 biology-13-00785-t002:** Half-maximal inhibitory concentrations (µg/mL) of selected *Moringa oleifera* leaf extracts in lipoxygenase, xanthine oxidase, and hyaluronidase inhibition assays.

Extracts (Samples)	IC50 (µg/mL)		
	Lipoxygenase Inhibition	Xanthine Oxidase Inhibition	Hyaluronidase Inhibition
TJM5	340.70 ± 9.08 ^c,d,m^	NA	424.33 ± 39.87 ^a,b,c,d,e,k^
TJM21	423.43 ± 4.36 ^h,j,m^	336.60 ± 174.52	633.65 ± 59.81 ^a,b,c,k^
TJM23	396.90 ± 0.55 ^a,g,m^	NA	196.10 ± 8.45 ^d^
TJM24	344.87 ± 3.73 ^c,d,e,m^	NA	292.32 ± 19.81 ^d,k^
TJM27	396.23 ± 14.74 ^a,b,g,m^	NA	319.99 ± 21.45 ^a,d,k^
TJM26	357.47 ± 14.73 ^m^	507.57 ± 361.40	299.62 ± 19.65 ^a,d^
TJM30	245.67 ± 2.42 ^m^	>1000.00	476.27 ± 90.79 ^a,b,c,e,k^
NSM16	340.80 ± 13.46 ^c,d,m^	>1000.00	108.96 ± 6.15
NSM22	401.93 ± 16.88 ^j,m^	450.73 ± 106.84	167.30 ± 14.61
NSM23	307.37 ±5.71 ^c,d,e,m^	417.70 ± 40.26	216.84 ± 7.65 ^k^
**Positive controls**			
NDGA	6.75 ± 0.58		
Allopurinol		1.00 ± 0.01	
Apigenin			100.59 ± 5.70

IC50 values were calculated via non-linear regression based on respective dose–response curves. The mean ± S.E.M. values are shown (*n* ≥ 3 independent experiments). NA indicates not active, while NT indicates not tested. The following letters mark significant differences (*p* < 0.05) in probability levels of each column, as assessed by ANOVA followed by Tukey’s multiple comparison test: (a) comparison with NSM16; (b) comparison with NSM23; (c) comparison with NSM22; (d) comparison with TJM21; (e) comparison with TJM23; (f) comparison with TJM24; (g) comparison with TJM5; (h) comparison with TJM28; (i) comparison with TJM 27; (j) comparison with TJM30; (k) comparison with apigenin; (m) comparison with NDGA; and (n) comparison with allopurinol.

**Table 3 biology-13-00785-t003:** Pearson’s correlation of astragalin with the anti-inflammatory activity of *Moringa oleifera* extracts.

Pearson Correlation	Hyaluronidase Inhibitory Activity (IC50)	Lipoxygenase Inhibitory Activity (IC50)	Xanthine Oxidase Inhibitory Activity (IC50)
r	−0.1839	0.08076	−0.5052
*p*-squared	0.03381	0.006522	0.2552
*p*-value (two-tailed)	0.6111	0.8245	0.1364
*p*-value summary	ns	ns	ns

Note: ns = not significant.

**Table 4 biology-13-00785-t004:** Vegetative growth traits of *Moringa oleifera* branch cuttings in different concentrations of indole-3-butyric acid (IBA) after 3 months under nursery conditions.

IBA Level (ppm)	Rooting (%)	Shooting (%)	Shoot No	Root No	Shoot Length (cm)	Root Length (cm)	Shoot Dry Weight (g)	Root Dry Weight (g)
0 (T0)	80.0 ± 5.77 ^a^	90.0 ± 4.08 ^a^	2.29 ± 0.29 ^a^	4.14 ± 1.16 ^a^	25.31 ± 5.35 ^ab^	8.86 ± 4.29 ^ab^	4.14 ± 0.91 ^ab^	1.00 ± 0.21 ^b^
1000 (T1)	80.0 ± 4.08 ^a^	95.0 ± 2.88 ^a^	3.17 ± 0.98 ^a^	3.67 ± 0.71 ^a^	19.97 ± 9.33 ^ab^	20.51 ± 7.66 ^a^	8.41 ± 1.42 ^a^	3.17 ± 0.44 ^a^
3000 (T2)	55.0 ± 6.46 ^b^	80.0 ± 2.40 ^b^	3.0 ± 0.62 ^a^	2.57 ± 0.69 ^a^	6.26 ± 1.15 ^b^	5.87 ± 1.92 ^b^	0.99 ± 0.36 ^b^	1.48 ± 0.43 ^b^
5000 (T3)	27.5 ± 7.50 ^c^	75.0 ± 2.88 ^b^	3.33 ± 0.49 ^a^	3.0 ± 1.13 ^a^	29.18 ± 7.84 ^a^	6.6 ± 2.73 ^ab^	4.23 ± 2.10 ^b^	0.73 ± 0.33 ^b^

Means followed by the same letters are not significantly different at *p* ≤ 0.05; ± indicates the standard error of the mean.

**Table 5 biology-13-00785-t005:** Vegetative growth traits of *Moringa oleifera* branch cuttings with varying diameters after 3 months under nursery conditions.

Diameter of Cuttings (cm)	Rooting Percentage (%)	Shoot Number	Root Number	Shoot Length (cm)	Root Length (cm)	Shoot Dry Weight (g)	Root Dry Weight (g)
2.5–≤3.5 (D0)	80.00 ± 5.77 ^a^	2.43 ± 0.30 ^b^	5.29 ± 1.11 ^ab^	33.14 ± 2.90 ^a^	10.66 ± 4.07 ^ab^	3.23 ± 0.78 ^c^	0.37 ± 0.04 ^b^
3.5–≤4.5 (D1)	56.67 ± 6.67 ^b^	3.57 ± 0.57 ^b^	5.42 ± 0.37 ^ab^	33.50 ± 3.92 ^a^	5.45 ± 0.56 ^b^	10.96 ± 1.75 ^b^	1.08 ± 0.21 ^b^
4.5–≤5.5 (D2)	53.33 ± 3.33 ^b^	3.33 ± 0.68 ^b^	4.50 ± 0.50 ^b^	27.08 ± 5.17 ^a^	15.11 ± 3.02 ^a^	13.23 ± 1.84 ^b^	2.46 ± 0.17 ^a^
5.5–≤6.5 (D3)	60.00 ± 5.77 ^b^	9.83 ± 0.77 ^a^	7.83 ± 0.44 ^a^	37.58 ± 3.77 ^a^	13.09 ± 0.79 ^ab^	24.47 ± 1.46 ^a^	3.69 ± 0.55 ^a^

The mean followed by the same letters is not significantly different at *p* ≤ 0.05; ± indicates the standard error of the mean.

**Table 6 biology-13-00785-t006:** Vegetative growth traits of *Moringa oleifera* branch cuttings with varying lengths after 3 months under nursery conditions.

Length of Cuttings (cm)	Rooting Percentage (%)	Shoot Number	Root Number	Shoot Length (cm)	Root Length (cm)	Shoot Dry Weight (g)	Root Dry Weight (g)
30 (L0)	70.00 ± 5.77 ^a^	2.50 ± 0.22 ^b^	3.20 ± 0.51 ^b^	34.00 ± 2.09 ^a^	9.20 ± 3.01 ^a^	3.26 ± 0.63 ^b^	0.34 ± 0.03 ^b^
40 (L1)	56.67 ± 6.66 ^a^	5.88 ± 1.39 ^a^	4.63 ± 0.92 ^ab^	22.15 ± 4.36 ^a^	3.05 ± 0.65 ^b^	10.68 ± 2.64 ^a^	1.16 ± 0.20 ^b^
50 (L2)	63.33 ± 3.33 ^a^	5.27 ± 0.40 ^a^	5.33 ± 0.67 ^ab^	31.90 ± 5.08 ^a^	5.01 ± 0.64 ^ab^	14.02 ± 2.71 ^a^	1.96 ± 0.15 ^ab^
60 (L3)	66.66 ± 3.33 ^a^	6.23 ± 0.62 ^a^	6.27 ± 0.53 ^a^	23.00 ± 2.56 ^a^	3.74 ± 0.56 ^ab^	18.00 ± 2.13 ^a^	3.24 ± 1.05 ^a^

The mean followed by the same letters is not significantly different at *p* ≤ 0.05; ± indicates the standard error of the mean.

**Table 7 biology-13-00785-t007:** Vegetative growth traits of *Moringa oleifera* branch cuttings in different types of growth substrates after 3 months under nursery conditions.

Rooting Substrates	Rooting Percentage (%)	Shoot Number	Shoot Length (cm)	Shoot Dry Weight (g)	Root Dry Weight (g)
M1	76.67 ± 3.33 ^a^	10.93 ± 1.03 ^a^	50.77 ± 4.92 ^a^	37.86 ± 4.48 ^a^	3.02 ± 0.55 ^a^
M2	66.67 ± 3.33 ^ab^	5.44 ± 0.52 ^b^	45.56 ± 6.33 ^bc^	25.30 ± 3.48 ^ab^	1.54 ± 0.28 ^ab^
M3	53.33 ± 3.33 ^b^	6.00 ± 0.95 ^b^	60.02 ± 9.71 ^a^	30.21 ± 3.72 ^a^	1.46 ± 0.32 ^ab^
M4	53.33 ± 3.33 ^b^	6.40 ± 0.74 ^b^	20.78 ± 3.06 ^b^	11.61 ± 1.94 ^b^	2.33 ± 0.35 ^a^
M5	56.67 ± 8.82 ^b^	5.43 ± 1.05 ^b^	21.04 ± 6.53 ^b^	12.89 ± 3.32 ^b^	0.65 ± 0.26 ^b^

The mean followed by the same letters is not significantly different at *p* ≤ 0.05; ± indicates the standard error of the mean.

## Data Availability

The data that support the findings of this study are contained within the article.

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
