# Peer review of "Crop Improvement of Moringa oleifera L. through Genotype Screening for the Development of Clonal Propagation Techniques of High-Yielding Clones in Malaysia"

_biology, 2024, doi:10.3390/biology13100785_

Round 1

Reviewer 1 Report

Comments and Suggestions for Authors

Moringa oleifera is a valuable tree species. In this manuscript, mother trees with good growth performance were collected in Malaysia. Genotypes with high astragalin content were selected and cutting propagation techniques were investigated. This study provided several high-yielding clones of Moringa oleifera for Malaysia.

Comments:

1. In title, this study should be limited in Malaysia.

2. Were the mother trees from natural population?

3. How many mother trees were selected totally?

4. It is necessary to conduct an analysis of geographical variation for the astragalin content.

5. The titles of figures should be below of the figures.

6. The authors should provide a figure to show the geographic distribution of the selected mother trees.

7. The correlationship between astragalin contents and the lipoxygenase, xanthine oxidase, and hyaluronidase inhibition assays should be analysed.

8. Subtitles in Results and Discussion part are necessary.

Reviewer 2 Report

Comments and Suggestions for Authors

The manuscript contains new findings in the field of gene resource evaluation and vegetative propagation of Moringa oleifera. The manuscript is well-prepared with an adequate number of graphs, tables, and photographic documentation.

Notes and Comments:

  • Line 314: What type of fertilizer and at what concentration was used?
  • Line 394: The description of the abbreviations for the three Moringa regions was omitted in Table 3.
  • Line 447 – Table 2: The full Latin name for Moringa should be provided.
  • Line 677: Could the health condition of the cuttings have affected their mortality, and was the occurrence of fungal diseases monitored?
